# Research on Pressure-Flow Characteristics of Pilot Stage in Jet Pipe Servo-Valve

**DOI:** 10.3390/s23010216

**Published:** 2022-12-25

**Authors:** Shuangqi Kang, Xiangdong Kong, Jin Zhang, Ranheng Du

**Affiliations:** 1School of Mechanical Engineering, Yanshan University, Qinhuangdao 066004, China; 2School of Naval Architecture and Ocean Engineering, Jiangsu Maritime Institute, Nanjing 211199, China

**Keywords:** jet pipe servo-valve, pilot stage, pressure-flow characteristics, impinging jet

## Abstract

Jet pipe servo-valves are widely used in high-precision servo control systems. However, the accuracy of pressure-flow characteristic equations relevant to its pilot stage needs to be improved. In contrast to the traditional analytical approach using the orifice equation, the article investigates the pressure and flow characteristics of the pilot stage based on the impact jet principle. Taking the pre-stage of a certain type of jet pipe servo-valve as an example, the flow field is simulated using ANASYS software. By comparing the simulation data with the calculation results, the pressure characteristic model is basically consistent with the simulation data, and the relative error is less than 2.3%. The error between the revised flow characteristic model and the simulation result is small, and the maximum deviation is less than 0.0022 L/min. Finally, to verify the applicability of the model to other specifications, the experimental data in the literature are compared with the theoretical calculation results, the maximum relative error of pressure characteristics is 2.19%, and the relative error of flow characteristics is less than 5.34%.

## 1. Introduction

The electro-hydraulic servo-valve, which is widely used in military, aerospace, industry and other fields, is an electronic hydraulic component that can precisely transform low-power electrical signals into high-power hydraulic signals (e.g., pressure, flow) [1]. The jet pipe servo-valve is a two-stage electro-hydraulic servo-valve with force feedback, as shown in Figure 1. Its pilot stage hydraulic amplifier is a jet pipe valve with a single nozzle to direct oil into the left and right receiving holes of the receiver. Under the action of electrical signals, the deflected nozzle causes a pressure difference between the left and right receiving holes. This causes the main spool to be moved. As a result, the servo-valve has a flow output proportional to the electrical signal [2]. The oil in the pre-stage has undergone the energy conversion process from pressure energy to kinetic energy and then back to pressure energy. The minimum channel size is large (on the order of 100 μm), and there is good resistance to pollution. In addition, there is only one nozzle in the front stage of the jet pipe servo-valve. If the nozzle is totally blocked, the main spool will automatically return to the center due to the equal pressure in the two receiving holes; however, partial blockage of the nozzle only reduces the performance of the servo valve without causing failure or runaway [3]. Therefore, the jet pipe servo-valve is mainly applied in jet engine hydraulic control systems and flight control systems [1,4]. In order to optimize the structural parameters and improve performance, there have been many research attempts addressing its design and analysis, especially that of the pilot stage of the servo-valve.

According to the flow continuity equation of the cavity at both ends of the main spool, Somashekhar et al. [5,6] obtained the Laplace transform of the recovery pressure of the pilot stage, and established the dynamics model of the symmetrical cylinder controlled by servo-valve. The stiffness values required for feedback spring assembly and spring tube assembly were obtained through finite element analysis and experimental verification. Further, the block diagram of a valve-controlled cylinder was established in Simulink for simulation research. In subsequent studies, Samashekhar [7] built the fluid-structure interaction finite element model of the jet pipe servo valve and investigated the velocity and displacement responses of both the jet pipe assembly and the main spool, together with the pressure changes of the control chamber at both ends of the main spool with the application of force at the armature. P. Zhao et al. [8] established the structural model of the armature component, the magnetic field model of the torque motor, and the flow field model between the jet pipe and the receiver, respectively, based on ANASYS software, and obtained the fitting relationship between the force and torque provided by the feedback spring, the electromagnetic torque output by the torque motor, and the pressure difference between the two ends of the spool. The dynamic characteristics of jet pipe servo-valve were studied. Wu et al. [9] accomplished the transient simulation experiments of the front stage of jet pipe servo-valve using an LES method based on ANASYS software, analyzed the effects of inlet pressure and outlet pressure on three types of cavitation and the pressure vibration induced by cavitation. In addition, the effect of wedge length on the flow field of the jet pipe valve was also analyzed. The cavitation phenomenon could be restrained with the increasing wedge length. But, when the length exceeded a certain value (0.03 mm), the energy loss increased sharply. However, most of these studies were focused on a certain type of servo valve, and pressure and flow characteristic equations suitable for the pilot stage have not been extracted.

Guo and Yin [10] divided the flow field of the jet pipe valve into two stages and obtained the recovery pressure formula of the left and right cavities of the receiver by using the Bernoulli equation and the continuity equation. Pham and Tran [11] divided the flow field of the pre-stage into three stages, and deduced the pressure and flow characteristic equations by applying the Bernoulli equation, small-hole flow equation of the nozzle, and flow continuity equation. The theoretical results were consistent with the experimental results in terms of the rule of change, but a correction factor was needed to add to the model of velocity according to the experiment. Yuesong Li [12] described the dynamic model of oil in the receiving holes as equivalent to the piston model, and deduced the pressure and flow characteristic equations of the jet pipe valve according to the free submerged jet theory, the small-hole flow formula, and the momentum theorem. Comparison between calculation results and simulation/experiment results showed that the pressure formula derived from the theory was reliable, and the flow formula was reliable at a small opening. Chen Jia [13] regarded the oil in the receiving hole as a moving liquid piston, and deduced the pressure and flow equations of the jet pipe assembly based on the small-hole flow formula, momentum theorem, and flow continuity equation under the premise of considering the reflected jet. At the same time, according to the theory of turbulent impingement jet and momentum theorem, the reverse impulse force of the receiving hole on the nozzle was analyzed. However, the working principle of oil injection from the nozzle is completely different from that of throttling, and the small-hole flow formula cannot be simply used to study jet velocity.

In fact, the flow in the pilot stage of the jet pipe servo-valve should be regarded as an impact jet, which is divided into the free jet region and the impingement region. After being ejected from the nozzle, the oil is shot into the stationary working medium and carries the idle fluid around in its motion. Due to the short jet distances involved, the jet beam strikes the upper surface of the receiver in the form of a potential core surrounded by an annular shear layer because it is not fully developed, which is different from the traditional impact jet. In this case, the assumption that the jet is emitted from a point source is not appropriate [14]. Meanwhile, because of the influence of the upper surface of the receiver and the wedge, the potential flow core ends in advance, and the impingement jet begins. Finally, the jet beam is split by the split tip and flows into the left and right receiving holes on the receiver respectively. Obviously, the pilot stage flow field should be studied and the pressure and flow characteristic equations should be established using the impact jet theory.

In Section 2, the calculation formula of the flow area in the elliptic receiver hole as a function of the nozzle hole position is given. The flow field of the pre-stage of the jet pipe is divided into three stages, and the pressure-flow model describing the pre-stage is established by the Bernoulli equation, impinge jet theory, and the momentum theorem. In Section 3, a certain type of jet pipe servo-valve is taken as an example to conduct a simulation study based on Fluent. Section 4 compares the theoretical calculation model with the simulation and revises the flow characteristic equation. In addition, in order to verify that the theoretical model proposed in the article is not only applicable to one size, calculations are carried out according to the key structural parameters given in the reference [12], and the values are compared with the experimental data. Section 5 gives the conclusion.

## 2. Mathematical Model

The structure diagram of the servo-valve of the jet pipe is shown in Figure 1. The oil with a certain pressure is transported to the jet pipe through a filter screen. When the input electrical signal from the coil on the armature assembly is zero (there is no signal to the torque motor), the pressure in the two receiving channels is equal because of the jet impacting on each receiving hole equally. So equal pressure is applied at the end of each spool, and the main spool remains in the middle position. When the input electrical signal of the coil is not zero, the jet pipe assembly deflects along with the armature around the pivot point, resulting in unequal jet inflow in the two receiving holes and unequal pressure in the two receiving channels. Therefore, the main spool moves under the action of differential pressure and the servo valve starts to work [15,16].

### 2.1. Flow Area

As the receiving hole of the receiver in Figure 1 is a cylindrical hole, the shape of the receiving hole on the surface of the receiver must be oval. The projection of the jet port and the receiving oil ports on the upper surface of the receiver is shown in Figure 2.

With the upper surface of the receiver regarded as an inclined plane in the spatial coordinate system, and the prime lines on the cylindrical hole surface regarded as a group of parallel lines, a plane perpendicular to the axis is made through any point on the cylindrical hole axis, as shown in Figure 3. According to the projection properties, the major and minor axes of the ellipse are $\;R_{r}/cos\theta_{r}$, $R_{r}$ respectively.

A Cartesian coordinate system is created with O_1_ as the origin in Figure 2. When the jet pipe shifts to the right by Δx, the flow area of the right receiving hole is shown in Figure 4.

Considering the geometric relationship of the figure shown in Figure 4, the flow area A_1_(*x*) is obtained:(1)$$A_{1}\left(x\right)=\left(S_{sector-O^{\prime }N_{1}N_{4}}+S_{sector-O_{1}N_{1}N_{4}}\right)-S_{\Delta O_{1}N_{1}N_{4}}-S_{\Delta O^{\prime }N_{1}N_{4}}$$
since N1 and N4 belong to both circle O’ and ellipse O1, that is to say, the coordinates of N1 and N4 satisfy the solution of the following equations:(2)$$\left\{\begin{array}{c} {\left(\frac{x}{R_{r}/cos\theta_{r}}\right)}^{2}+{\left(\frac{y}{R_{r}}\right)}^{2}=1\\ {\left(x+\frac{R_{r}}{cos\theta_{r}}+\frac{e}{2}-\Delta x\right)}^{2}+y^{2}=R_{j}^{2} \end{array}\right.$$

Assuming that the real number solutions satisfying equation group 2 are ($-x_{N1}$,$y_{N1}$) and ($-x_{N1}$,$-y_{N1}$), then:(3)$$∠N_{1}O_{1}N_{4}=\alpha_{1}=2arctan\frac{y_{N1}}{x_{N1}}$$
(4)$$∠N_{4}ON_{1}=\alpha_{4}=2arctan\frac{y_{N1}}{R_{r}/cos\theta_{r}+e/2-\Delta x-x_{N1}}$$

Considering the small difference in length between the long half axis and the short half axis of the ellipse, for easy calculation, the sector-shaped area of the ellipse is approximated, taken as
(5)$$S_{sector-O_{1}N_{1}N_{4}}\approx \frac{1}{cos\theta_{r}}R_{r}^{2}\alpha_{1}$$

Based on the triangular area formula, circular sector area formula and elliptic sector area formula, we have
(6)$$A_{1}\left(x\right)=\frac{\frac{1}{cos\theta_{r}}R_{r}^{2}\alpha_{1}+R_{j}^{2}\alpha_{4}-(2*x_{N1}y_{N1}+R_{j}^{2}sin\alpha_{4})}{2}$$

Similarly, a Cartesian coordinate system is created with O_2_ as the origin, where the coordinates of N_2_ and N_3_ satisfy the following equations:(7)$$\left\{\begin{array}{c} {\left(\frac{x}{R_{r}/cos\theta }\right)}^{2}+{\left(\frac{y}{R_{r}}\right)}^{2}=1\\ {\left(x-\frac{R_{r}}{cos\theta }+\frac{e}{2}-\Delta x\right)}^{2}+y^{2}=R_{j}^{2} \end{array}\right.$$

Let the real number solutions satisfying equation set 7 be ($x_{N2}$,$y_{N2}$) and ($x_{N2}$,$-y_{N2}$), then:(8)$$∠N_{1}O_{1}N_{4}=\alpha_{2}=2arctan\frac{y_{N2}}{x_{N2}}$$
(9)$$∠N_{4}ON_{1}=\alpha_{3}=2arctan\frac{y_{N2}}{R_{r}/cos\theta_{r}+e/2+\Delta x-x_{N2}}$$
So we have:(10)$$A_{2}\left(x\right)=\frac{\frac{1}{cos\theta_{r}}R_{r}^{2}\alpha_{2}+R_{j}^{2}\alpha_{3}-(R_{r}^{2}sin\alpha_{2}+R_{j}^{2}sin\alpha_{3})}{2}$$

### 2.2. Pressure and Flow Characteristics Model

Assuming that the fluid is incompressible, the oil dynamic process in the pilot stage can be equivalent to three stages, as shown in Figure 5. The first stage is the flow of oil in the pipe from section I to section II. The second stage is the oil impingement jet, which can be divided into two regions according to the jet theory. One is the free submerged jet region, from section II to section III; the other is the impact zone, from section III to section IV. The third stage is the flow of oil in the receiving holes, from section IV to section Ⅴ.

#### 2.2.1. Pipe Flow

In the first stage, the oil supply pressure from section I into the jet pipe is $p_{s}$. When the oil reaches section II, the pressure drops to the return pressure $p_{t}$, and the velocity changes to $v_{j}$. Considering the high machining precision of the jet pipe the tolerance grade of which is IT5~IT6, the loss of oil along the jet pipe is ignored. According to Bernoulli’s equation [17], we have:(11)$$h_{s}+\frac{p_{s}}{\rho g}+\frac{v_{s}^{2}}{2g}=h_{j}+\frac{p_{t}}{\rho g}+\frac{v_{j}^{2}}{2g}$$
where:


(1)$h_{s}$
—The gravitational potential energy of oil at the inlet of the jet pipe;(2)ρ—Oil density, $\mathrm{kg}/m^{3}$;(3)$p_{s}$—Inlet pressure, MPa;(4)$v_{s}$—The velocity of the oil at the inlet of the jet pipe, approximately considered to be zero;(5)$h_{j}$—The gravitational potential energy of the oil at the nozzle mouth;(6)$p_{t}$—The return pressure, MPa;(7)$v_{j}$—The velocity of the oil at the nozzle mouth, m/s.


The velocity of the oil at the nozzle outlet can be obtained:(12)$$v_{j}=\sqrt{\frac{2}{\rho }(p_{s}-p_{t})+2g\left(h_{s}-h_{j}\right)}$$

#### 2.2.2. Impact Jet

In the second stage, the oil is injected into the surrounding static medium from the nozzle and carries the static liquid to move with it. Therefore, the jet velocity will gradually decrease, the jet width will gradually increase, and the potential nucleus will gradually disappear. Normally, the length of the potential core is approximately six times the diameter of the jet hole. The usual research on an impinging jet is based on the full development of free-submerged jet [14]. However, in the jet pipe valve, the distance between the nozzle and the upper surface of the receiver is shorter than the length of the potential core, so the calculation formula of the free jet zone and impact jet zone cannot simply be applied.

The jet in the free submerged jet region from section II to section III is in the initial stage, so it is approximately considered that the potential core area is equal to the jet hole area. According to the theory of a free submerged jet, the velocity of oil in the potential core is equal everywhere. It is approximately considered that the velocity in section III is equal to the jet velocity of oil $v_{j}$.

In the impact zone, the jet beam is divided into two parts by the split tip and flows into the left and right receiving holes, respectively. Since the wall velocity is zero, the tip of the wedge is regarded as the stagnation point of the impact zone. According to Bernoulli’s equation [17], we have
(13)$$p_{c}=p_{t}+\rho g\left(L_{j}-x_{0}\right)+\frac{1}{2}\rho v_{j}^{2}p_{c}=p_{t}+\rho g\left(L_{j}-x_{0}\right)+\frac{1}{2}\rho v_{j}^{2}$$
where:
(1)$p_{c}$—Stagnation pressure, MPa;(2)$L_{j}$—Vertical distance from the nozzle to receiving hole, m;(3)$x_{0}$—The size of the free jet zone, i.e., the distance between sections II and III, m.


According to references [18,19], it can be known that:(14)$$x_{0}=0.86L_{j}$$

Combining Formulas (12)–(14), the result is obtained
(15)$$p_{c}=p_{s}+0.14\rho L_{j}g+\rho g\left(h_{s}-h_{j}\right)$$

In section IV, the pressure outside the two receiving holes can be approximately regarded as the return oil pressure $p_{t}$. Assuming no energy loss, the velocity is the maximum, which is
(16)$$v_{c}=\sqrt{\frac{2}{\rho }p_{c}}$$

For simple calculation, the average pressure and average velocity [14] in the two receiving holes in section IV are
(17)$$\bar{p_{c}}=\left(1.08-1.14\frac{R_{j}}{2R_{j}}\right)p_{c}=0.51p_{c}$$
(18)$$\bar{v_{c}}=0.3756v_{c}$$

#### 2.2.3. Flow in the Receiving Holes

In the third stage, the oil entering the right receiving hole in time interval *dt* is regarded as the research object [12], as shown in Figure 6. The equivalent oil mass of the flow beam shot into the right receiving hole along the axis direction in time interval *dt* is:(19)$$dm_{j}=\rho A_{1}\left(\bar{v_{c}}cos\theta_{r}-v_{1r}\right)dt$$
where: $\bar{v_{c}}$—The average velocity of the equivalent beam at the upper surface of the receiver, m/s; and $v_{1r}$—Oil velocity in the right receiving hole, m/s.

According to the law of momentum:(20)$$F_{1r}=\left[\bar{p_{c}}A_{1}+dm_{j}g+p_{t}\left(A_{r}-A_{1}\right)\right]cos\theta_{r}-\frac{dm_{j}(\bar{v_{c}}cos\theta_{r}-v_{1r)}}{dt}$$
where: $F_{1r}$—The force exerted by the pressure of the right receiving hole on the equivalent moving piston of the right receiving hole, N; and $\bar{p_{c}}$—The average pressure of the equivalent flow beam at the upper surface of the receiver, MPa.

Substituting Formula (19) into Formula (20) and ignoring the influence of oil gravity and back pressure, then:(21)$$F_{1r}=\bar{p_{c}}A_{1}cos\theta_{r}-\rho A_{1}{\left(\bar{v_{c}}cos\theta_{r}-v_{1r}\right)}^{2}$$

Thus, the pressure caused by the flow beam in the right receiving hole is:(22)$$p_{1r}=\frac{F_{1j}}{A_{r}}=\bar{p_{c}}\frac{A_{1}}{A_{r}}cos\theta_{r}-\rho \frac{A_{1}}{A_{r}}{\left(\bar{v_{c}}cos\theta_{r}-v_{1r}\right)}^{2}v$$
where: $p_{1r}$—Pressure at right receiving mouth, MPa.


**The pressure characteristic model**


The difference between the recovery pressures of the two receiving holes is defined as the load pressure of the pre-stage. The pressure characteristic refers to the relationship between load pressure and the displacement of the jet pipe when the load is cut off [20]. Then, the oil flows into the left receiving channel as well. We have:(23)$$F_{2r}=\bar{p_{c}}A_{2}cos\theta_{r}+\rho A_{2}{\left(\bar{v_{c}}cos\theta_{r}-v_{2r}\right)}^{2}$$
(24)$$p_{2r}=\frac{F_{2j}}{A_{r}}=\bar{p_{c}}\frac{A_{2}}{A_{r}}cos\theta_{r}+\rho \frac{A_{2}}{A_{r}}{\left(\bar{v_{c}}cos\theta_{r}-v_{2r}\right)}^{2}$$

Thus, the load pressure between the two receiving channels is
(25)$$p_{L}=\bar{p_{c}}\frac{A_{1}-A_{2}}{A_{r}}cos\theta_{r}+\rho {\bar{v_{c}}}^{2}\frac{A_{1}-A_{2}}{A_{r}}cos^{2}\theta_{r}+\rho \frac{A_{1}-A_{2}}{A_{r}}\left[v_{1r}^{2}-v_{2r}^{2}-2\bar{v_{c}}cos\theta_{r}\left(v_{1r}-v_{2r}\right)v\right]$$

Cutting off the load means $q_{L}$ is zero, which is to say, $v_{1r}$ and $v_{2r}$ are both zero, and the pressure characteristic model of the flow field can be obtained:(26)$$p_{L}=\frac{A_{1}-A_{2}}{A_{r}}\bar{p_{c}}cos\theta_{r}+\rho {\bar{v_{c}}}^{2}\frac{A_{1}-A_{2}}{A_{r}}cos^{2}\theta_{r}$$


**The flow characteristic model**


The recovery flow of the receiving holes is defined as the load flow. The flow characteristic refers to the relationship between load flow and the displacement of the jet pipe when the two receiving channels are connected [20]. The oil would then flow in the opposite direction in the left receiving channel. We have:(27)$$F_{2r}=\bar{p_{c}}A_{2}cos\theta_{r}+\rho A_{2}{\left(\bar{v_{c}}cos\theta_{r}+v_{2r}\right)}^{2}$$
(28)$$p_{2r}=\frac{F_{2j}}{A_{r}}=\bar{p_{c}}\frac{A_{2}}{A_{r}}cos\theta_{r}+\rho \frac{A_{2}}{A_{r}}{\left(\bar{v_{c}}cos\theta_{r}+v_{2r}\right)}^{2}$$

The load flow is:(29)$$q_{L}=\frac{q_{1r}+q_{2r}}{2}=\frac{v_{1r}+v_{2r}}{2}A_{r}$$

According to Formula (22) and Formula (28), it can be obtained that:(30)$$v_{1r}=\bar{v_{c}}cos\theta_{r}-\sqrt{\frac{1}{\rho }}\sqrt{\frac{A_{r}}{A_{1}}p_{1r}-\bar{p_{c}}cos\theta_{r}}$$
(31)$$v_{2r}=-\bar{v_{c}}cos\theta_{r}+\sqrt{\frac{1}{\rho }}\sqrt{\frac{A_{r}}{A_{2}}p_{2r}-\bar{p_{c}}cos\theta_{r}}$$

Substituting Formulas (30) and (31) into Formula (29) results in:(32)$$q_{L}=\frac{A_{r}}{2}\sqrt{\frac{1}{\rho }}(\sqrt{p_{2r}\frac{A_{r}}{A_{2}}-\bar{p_{c}}cos\theta_{r}}-\sqrt{p_{1r}\frac{A_{r}}{A_{1}}-\bar{p_{c}}cos\theta_{r}})$$

Considering that the two receiving channels are connected with the jet area at this time, it is approximately believed that the pressure of the receiving channels and the jet area is consistent, namely $p_{1r}=p_{2r}=\bar{p_{c}}$. Therefore, Formula (33) is:(33)$$q_{L}=\frac{A_{r}}{2}\sqrt{\frac{1}{\rho }}(\sqrt{\bar{p_{c}}cos\theta_{r}-p_{1r}\frac{A_{r}}{A_{1}}}-\sqrt{\bar{p_{c}}cos\theta_{r}-p_{2r}\frac{A_{r}}{A_{2}}})$$

## 3. Simulation Research

### 3.1. Simulation Pre-Processing

#### 3.1.1. Geometric Model

The front-stage 3D model is shown in Figure 7a. In order to improve the efficiency and quality of mesh division and reduce the distortion of mesh, all curved surfaces will be ignored. The 3D model of the flow passage in the jet pipe valve is simplified as shown in Figure 7b. Key structural dimensions are shown in Table 1.

#### 3.1.2. Model Selection and Boundary Conditions Setup


**Model selection**


Enable the *Energy Equation* in the energy model.

In the viscosity model, we select *Realizable k-*ε for the turbulence model, *Enhanced Wall Fn* for near-wall treatment, and *Viscous Heating* to consider the temperature and *viscosity effect* more realistically [21].


**Material parameters**


The material parameters are shown in Table 2, and the oil viscosity is defined by the user-defined function UDF. The curve of viscosity changes with temperature as shown in Figure 8.


**Boundary conditions**


Boundary conditions are set as shown in Figure 9. Figure 9a is the model to solve pressure change and Figure 9b is the model to solve flow change.

#### 3.1.3. Meshing

As an approximate solution method, the accuracy of numerical simulation results is affected by many factors, such as mesh quality and mesh quantity. It is generally believed that the more the number of grids, the higher the accuracy of the solution. At the same time, the rapid increase in computing scale and storage space leads to a decrease in computing efficiency. Therefore, it is necessary to select a reasonable number of grids to improve the computational efficiency on the premise of ensuring the computational results [22].

According to the analysis in Section 2, the impact jet section of oil in the flow field of the pre-stage is very important, so the relevant area adopts an encrypted mesh, as shown in Figure 10a. The flow at other positions, such as the oil inlet end, oil return end, and the remote end of the receiving channel, is not violent, so the grid size adopts the system default, as shown in Figure 10b. The final pressure model has 237,521 grids, and the flow model has 3,275,055 grids.

### 3.2. The Simulation Results

Streamline distribution in the flow field is shown in Figure 11a. It can be seen from the figure that the oil flows in four directions after it exits the nozzle. One part is similar to the plate impact jet flow, directly along the upper surface of the receiver back to the return oil port. Another part of the oil flows into the receiving channel and then spills out, flowing back to the return oil port along the upper surface of the receiver. Most of the oil forms the vortex ring as shown in Figure 11b in the area from section II to section III, and diffuses in the surrounding space. The last part of the oil flows into the receiving channels, forming a ring flow. In the simulation analysis, most of the jet flows back to the oil tank, which is consistent with the large null flow of the actual jet pipe valve, indicating that the simulation model has certain accuracy.

The pressure distribution in the pre-stage flow field is shown in Figure 12. As can be seen from the figure, in the pipeline flow (as section I to section II in Figure 5), the oil pressure starts to decrease after the vena contracta and drops to $P_{t}$ at the nozzle. In the impact jet region (as section II to section IV in Figure 5), the oil pressure increases at the position above the upper surface of the receiver due to the impingement zone. The negative pressure area around the receiving hole is caused by the vortex dissipating part of the jet energy [23]. Therefore, compared with the left side, the negative pressure area on the right side has a lower pressure value and a larger area. When the oil enters the receiving channels, the pressure remains stable. During the flow from the nozzle to the receiving channels, the oil has completed two pressure changes, undergone the energy conversion from pressure energy to kinetic energy, and kinetic energy to pressure energy.

## 4. Results and Discussion

### 4.1. Comparison between Simulation and Theory

For convenient analysis, the offset interval is set to 5 μm. The parameters given in Table 1 were substituted into Formulas (2)–(10), and the curve of receiving area changing with offset value was obtained as shown in Figure 13. In this paper, only the right offset of the jet pipe is considered, and the left offset results are similar.

The pressure and flow at different offset values calculated according to the theoretical model, and corresponding characteristics obtained according to CFD simulation are shown in Figure 14.

It can be seen from Figure 14a that the calculated data of the pressure characteristic model match the simulation results very well, and the maximum relative error is 2.3%. This shows that the load pressure characteristic model based on impingement jet theory, namely Formulas (17) and (27), is relatively accurate.

It can be seen from Figure 14b that when the offset value of the jet pipe is small, the theoretical calculation curve of load flow is completely different from the simulation curve, and the variation trend is completely different. As the theoretical curve is always below the simulation curve during the whole process of the shift change, the flow characteristic equation (namely Formula (35)) is inaccurate. In other words, the pressure of the receiving channels and the jet area is unequal. Rather, the pressure of the receiving channels is less than the average pressure in the jet area. In fact, it is not appropriate to simply use dynamic analysis, as can be seen in Figure 11; the flow field of the jet pipe valve is very complex, and the flow pattern near the receiving hole includes plate impingement jet, oblique impingement jet, sputtering, vortex ring, etc. Not only does the oil return instead of flowing into the receiving holes, the diffusion oil after vortex ring movement may also flow into the receiving hole; it is not possible to simply set the receiving channel’s pressure.

According to the mass conservation theorem and the flow direction analysis in Section 3.2, in the flow field, most of the oil injected from the nozzle flows back to the tank through the oil return port, and the ring flow is due to flow area difference caused by the offset of the jet pipe, so we have:(34)$$Q_{L}=\left|A_{1}-A_{2}\right|\times \bar{v}\times cos\theta_{r}$$

In addition, considering the sputtering phenomenon and viscous friction loss in practice, the flow characteristic equation is
(35)$$Q_{L}=C_{d}\left|A_{1}-A_{2}\right|\times \bar{v}\times cos\theta_{r}$$
where: $C_{d}$—the flow coefficient. 

According to simultaneous Formulas (16), (18), and (35), and by substituting corresponding parameters in them, the load flow value changing with the offset was obtained. The comparison with the load flow obtained by CFD simulation is shown in Figure 15.

It can be seen from the figure that the changing trend of the revised flow characteristic model and the simulation result are the same, and the numerical error is small. The maximum value is 0.0022 L/min only when the offset value is 0.095 mm. This shows that the revised flow characteristics model is more consistent with the actual flow situation. 

### 4.2. Experiments

In order to verify the accuracy of the pressure and flow characteristic model established in this paper, the experimental results of the pre-stage of the jet pipe servo-valve given in references [12] and [20] are selected. The parameters are shown in Table 3, and the experimental data are shown in Table 4.

As before, only the case where the jet pipe is shifted to the right is analyzed. According to parameters shown in Table 3, the curve of the through-flow area changing with the offset value is drawn as shown in Figure 16. If e = 0.1 mm, as described in reference [12], then when the displacement is 1.0 mm, the projection of the jet hole will exceed the area range of the right receiving port, and the measured pressure and flow rate should be lower than those of 0.8 mm. Therefore, this paper considers that e = 0 mm. In addition, when the offset is 0.8 mm and 1.0 mm, the jet port is completely projected into the right receiving hole, which is not consistent with the actual operation, and these two points are not considered in this paper.

The pressure characteristic model (Formula (26)) and experimental results are shown in Figure 17. The theoretical pressure characteristic curve is close to the experimental one, and the maximum relative error of the pressure characteristic is 3.12%. In contrast, the maximum relative error was 5.48% in the reference [12]. It is worth noting that the average pressure is calculated using formula 17. However, when calculating the average velocity, the coefficient is 0.65 instead of 0.3756 (Formula (18)). This indicates that the velocity distribution of impinging jet in the experiment is not Gaussian distribution.

Figure 18 shows the comparison between the flow characteristic model (Formula (35)) and the experimental value, and the relative error is less than 5.34%, while the maximum relative error was 6.07% in reference [12]. It should be noted that the flow factor used in the calculation is 1. The error may be caused by a small calculated average velocity or almost no loss due to the large structure size and low oil inlet pressure. These items of work need further experimental study.

The jet pipe servo-valve used in the experiment has large size, large offset interval, small inlet pressure, and the distance between the nozzle and the upper surface of the receiver is unknown, which is very different from the jet pipe servo-valve studied earlier in this paper. Therefore, the average velocity coefficients needed for calculation of the two are inconsistent. This also indicates that for jet pipe servo-valves with different structure sizes, the shapes of the impingement jet are also different, and the calculations of the average pressure and velocity, and even the flow coefficient, may be different. How these calculated parameters vary with size structure requires further theoretical and experimental studies.

## 5. Conclusions

The flow pattern in the hydraulic amplifier of the jet pipe servo-valve is complex. There are three paths of the oil ejected from the nozzle to return to the oil tank. Some oil flows along the surface of the receiver, like the plate impact jet. Part of the oil entering the receiving holes spills out and flows to the return oil port via the receiver’s upper surface. A vortex ring is formed above the receiver upper surface, then it diffuses to surrounding space, and finally flows to the return oil port or the receiving holes.Near the receiving holes, oil flow includes plate impingement jet, oblique impingement jet, vortex ring, and sputtering. In particular, the latter two flow patterns are rarely involved in traditional analysis. This suggests that the reasonability of using the orifice flow formula to analyze and calculate the load flow of the hydraulic amplifier is still to be discussed.The calculation results of the pressure characteristic model agree well with the simulation results; this indicates that the pressure characteristic equation has certain accuracy. The calculation results of the modified flow characteristic model are in good agreement with the simulation results; this indicates that the load flow characteristic equation has certain accuracy.The experimental data mentioned in references [12] and [20] are used to verify the accuracy of pressure and flow characteristics models. The experimental data agree well with the theoretical data without considering the limit. It shows that the theoretical model can describe the pressure and flow characteristics of jet pipe valve. However, the selection of average pressure, average velocity and even amplification coefficient in the model should be determined according to the actual structural parameters of jet pipe servo-valve.Based on impingement jet theory, the flow and pressure characteristics of the pre-stage of the jet pipe servo-valve are studied. Since the flow field information in the impact jet depends on the structure sizes of the jet pipe valve, that is to say, different sizes may lead to different jet distribution in the impinging section. Then the method of determining the average pressure, average flow rate, and even the flow factor will need further research and experiment.

## Figures and Tables

**Figure 1 sensors-23-00216-f001:**
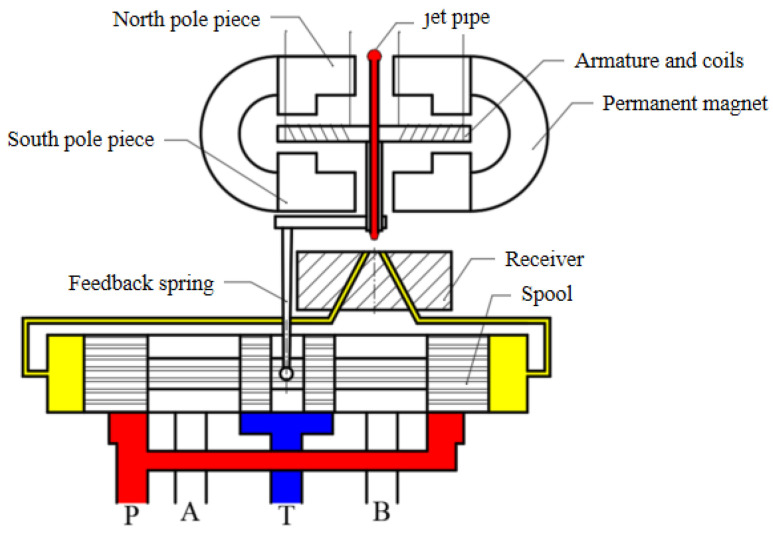
Schematic of jet pipe servo-valve. Red represents high pressure oil, yellow represents working oil, and blue represents oil flowing back to the tank.

**Figure 2 sensors-23-00216-f002:**
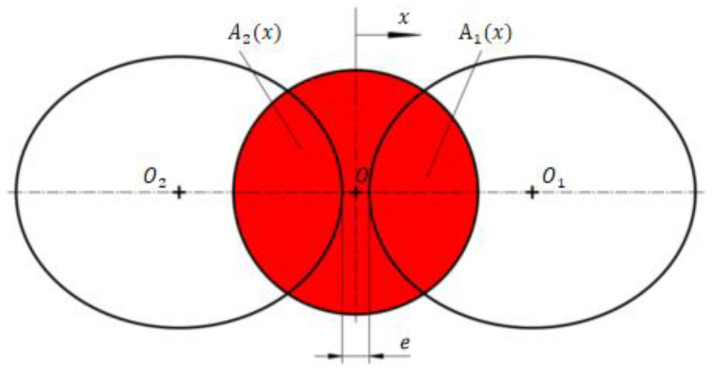
Projection of the upper surface of the receiver.

**Figure 3 sensors-23-00216-f003:**
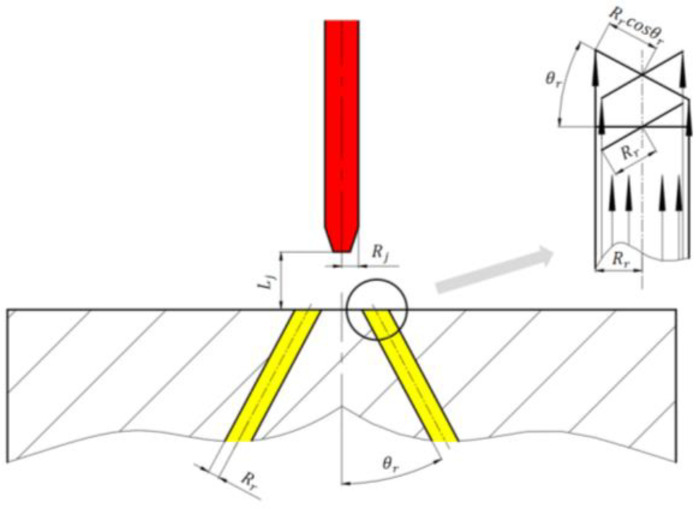
Projection relationship between the hole and the upper surface of the receiver.

**Figure 4 sensors-23-00216-f004:**
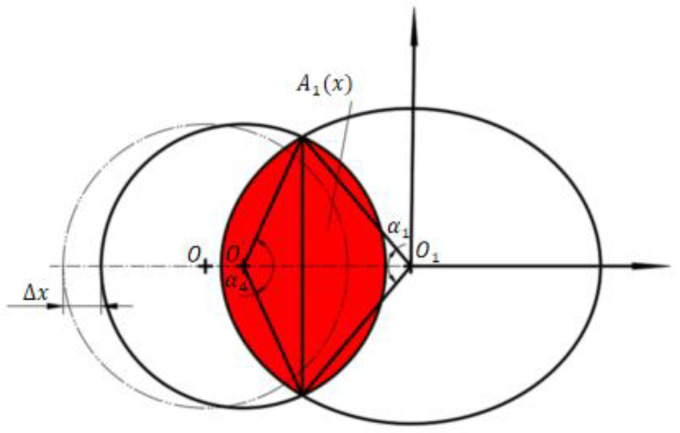
Schematic diagram of flow area of right receiving hole.

**Figure 5 sensors-23-00216-f005:**
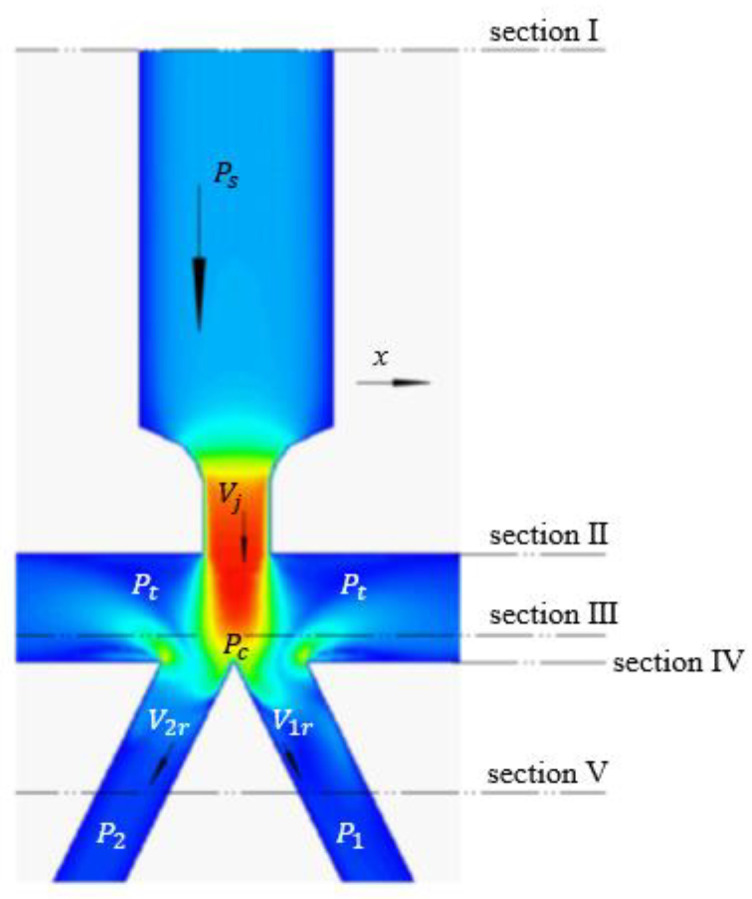
Velocity structure of the pilot stage.

**Figure 6 sensors-23-00216-f006:**
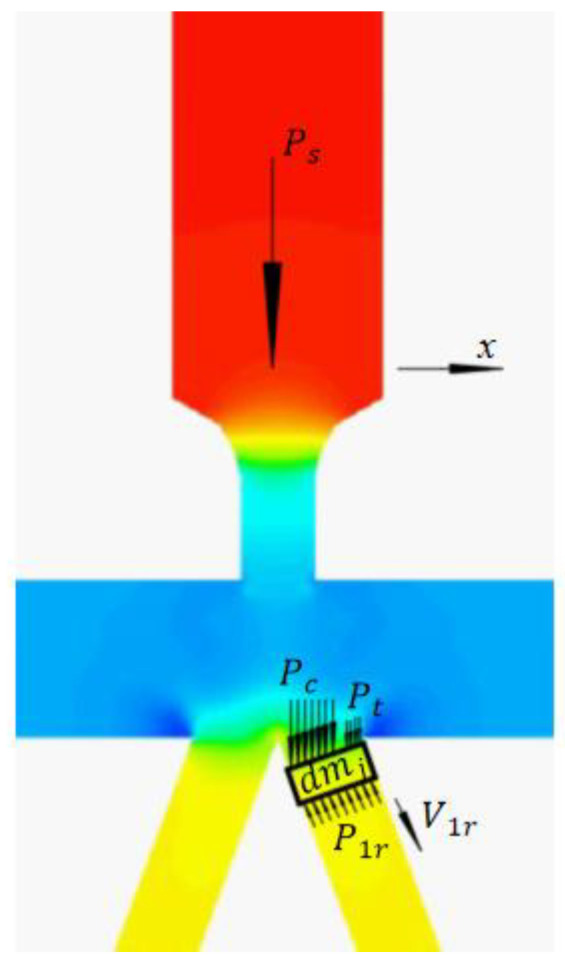
Stress analysis diagram.

**Figure 7 sensors-23-00216-f007:**
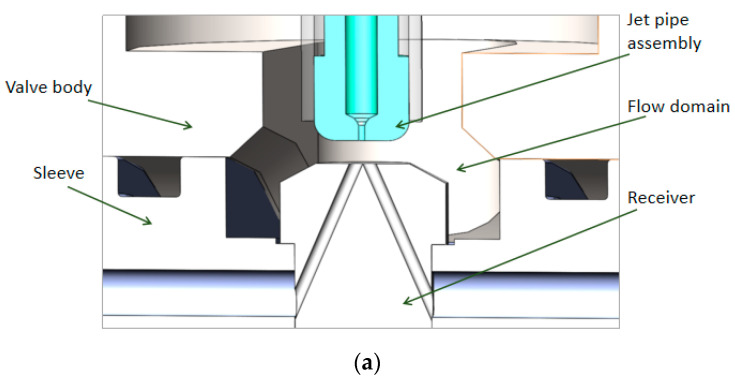
Pre-stage flow field model: (**a**) 3D model; and (**b**) flow passage DM model.

**Figure 8 sensors-23-00216-f008:**
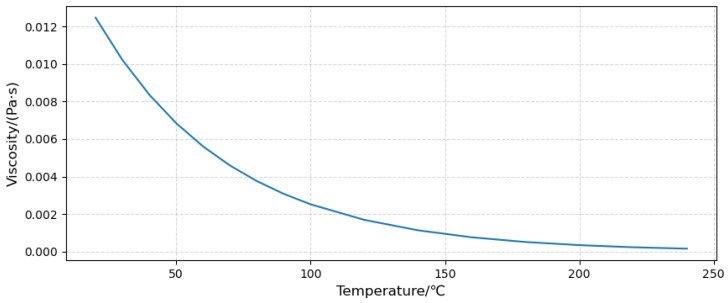
Viscosity curve with temperature.

**Figure 9 sensors-23-00216-f009:**
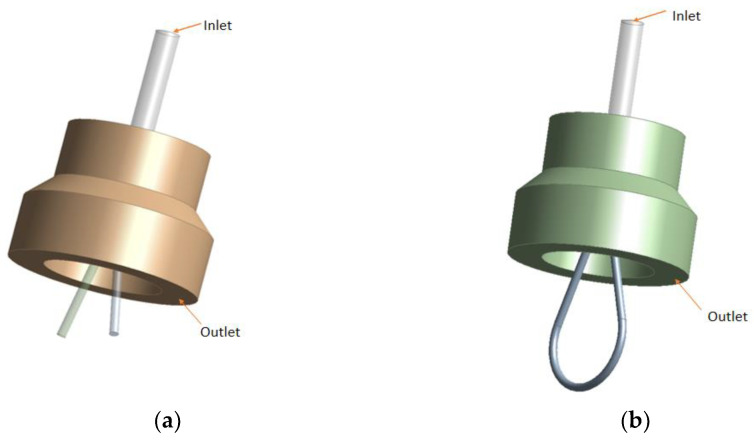
Schematic diagram of setting boundary conditions: (**a**) Block-load model; and (**b**) Connected model. Inlet: adopt *pressure inlet*, pressure $p_{s}$ is 21 MPa, and inlet oil temperature is 300 K. Outlet: *pressure outlet* is used. The oil return pressure $p_{t}$ is set to 0.5 MPa. Wall surface: slip-free fixed-wall surface.

**Figure 10 sensors-23-00216-f010:**
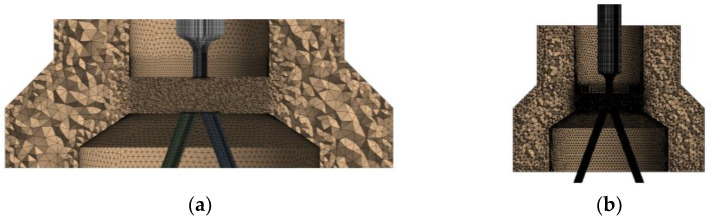
Grid model diagram: (**a**) partial encryption; and (**b**) the whole grid.

**Figure 11 sensors-23-00216-f011:**
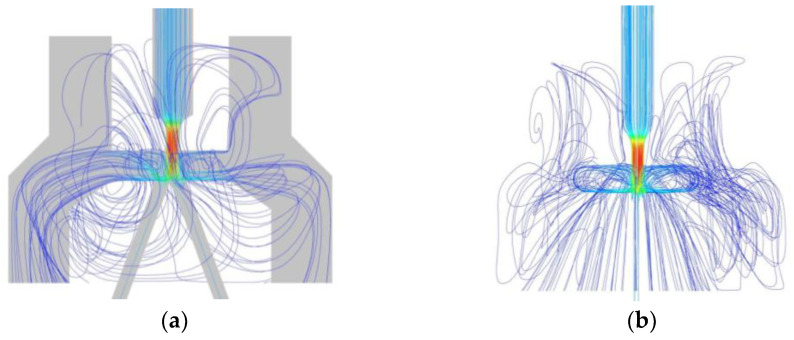
Streamline distribution diagram of jet tube migration. (**a**) Z projection; (**b**) X projection.

**Figure 12 sensors-23-00216-f012:**
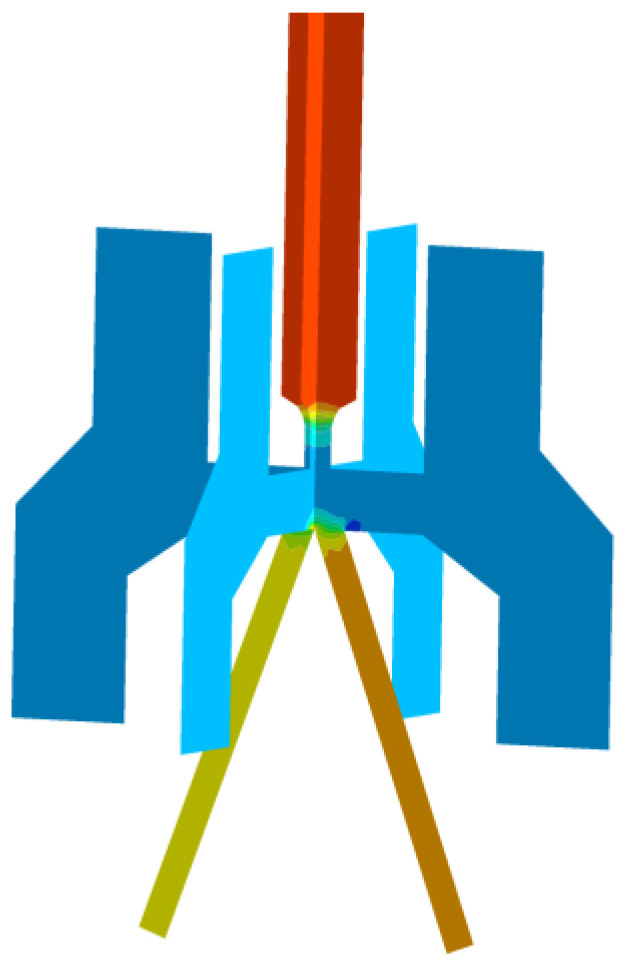
Pressure distribution diagram of jet pipe offset.

**Figure 13 sensors-23-00216-f013:**
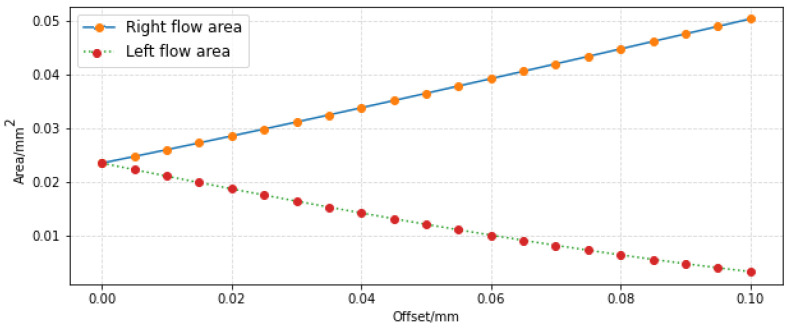
Variation curves of through-flow area.

**Figure 14 sensors-23-00216-f014:**
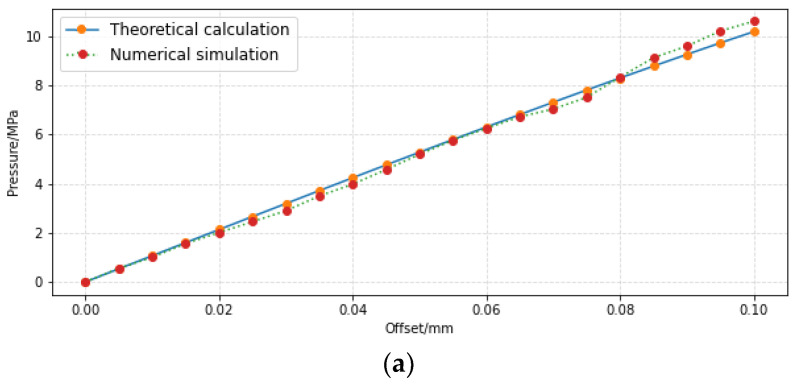
Calculation results versus simulations: (**a**) curves of pressure with offset; and (**b**) curves of flow with offset.

**Figure 15 sensors-23-00216-f015:**
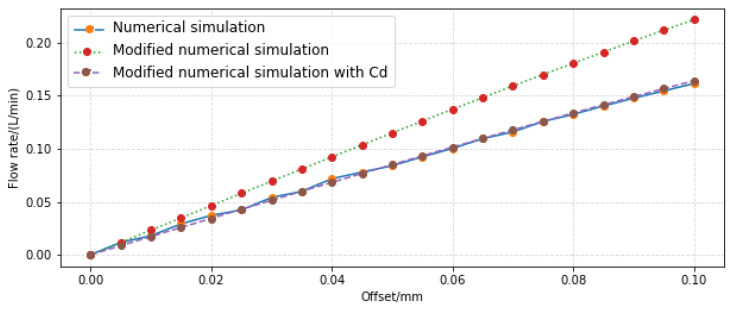
Variation curve of load flow with offset.

**Figure 16 sensors-23-00216-f016:**
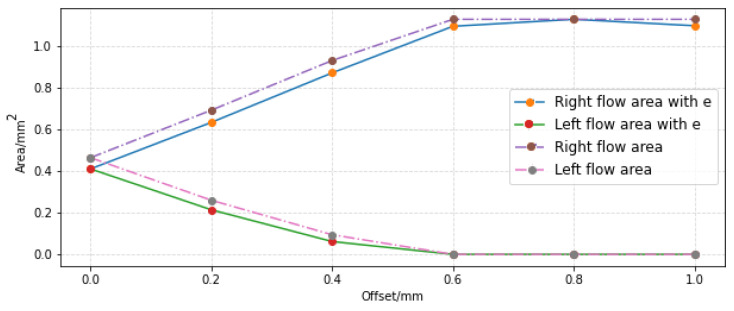
Variation curve of flow area with offset.

**Figure 17 sensors-23-00216-f017:**
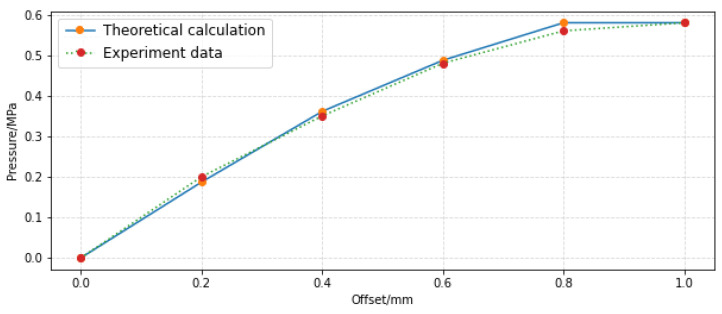
Comparison between theoretical pressure model and experiment.

**Figure 18 sensors-23-00216-f018:**
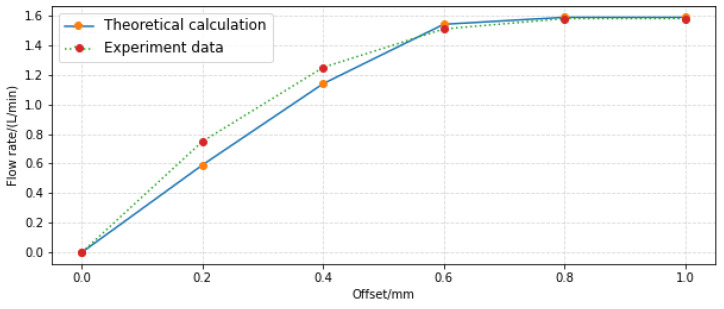
Comparison between theoretical flow model and experiment.

**Table 1 sensors-23-00216-t001:** Key structural parameter table.

Parameter	Symbol	Value	Unit
Nozzle radius	$R_{j}$	0.12	mm
Radius of receiving hole	$R_{r}$	0.14	mm
Injection spacing	$L_{j}$	0.5	mm
Angle of receiving hole	$\theta_{r}$	18	°
Spacing of receiving hole	e	0	mm

**Table 2 sensors-23-00216-t002:** Hydraulic oil parameters.

Project	Value	Unit
Density	850	$\mathrm{kg}/m^{3}$
Specific heat capacity at constant pressure	1940	J/kg·K
Coefficient of thermal conductivity	0.123	W/m·K

**Table 3 sensors-23-00216-t003:** Parameters value of a jet pipe valve.

Parameter	Symbol	Value	Unit
Nozzle radius	$R_{j}$	0.6	mm
Radius of receiving hole	$R_{r}$	0.75	mm
Supply pressure	$P_{s}$	0.6	MPa
Angle of receiving hole	$\theta_{r}$	15	°
Spacing of receiving hole	e	0.1	mm

**Table 4 sensors-23-00216-t004:** Experiment data.

Offset (mm)	Flow Rate (L/min)	Pressure (MPa)
0	0	0
0.2	0.75	0.2
0.4	1.25	0.35
0.6	1.51	0.48
0.8	1.58	0.56
1.0	1.75	0.58

## Data Availability

Not applicable.

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
