# Peer review of "Research on Pressure-Flow Characteristics of Pilot Stage in Jet Pipe Servo-Valve"

_sensors, 2022, doi:10.3390/s23010216_

Round 1
Reviewer 1 Report
The article investigates the pressure-flow characteristics of the pilot stage in jet pipe servo-valve, proposes pressure and flow characteristic calculation models based on the impact jet theory. The calculated results match with the simulation results of the flow field. And then the accuracy of the calculation models are verified by comparing with the experimental data.
My comments are as follows:
1) Contribution of the work should be clearly stated in the introduction.
2) There are many equations in section 2.2, are there any references to them?
3) Some comparisons should be added to show the advantage between the proposed method and the existing works.
4) The English of the paper should be improved.
5) In line 374, a space is missing.
6) In line426, a space is missing.
Reviewer 2 Report
This paper investigates the pressure and flow characteristics of a jet pipe servo-valve based on the impact jet principle.
Research topics could be interesting, because of widely use of electrohydraulic servo systems in modern industrial practice, military applications and aerospace, so studies oriented at achieving improved dynamic behavior of hydraulic systems are highly desirable. In view of this, the proposed work towards developing a precise mathematical model of the servo valve is particularly valuable. The manuscript offers an interesting and promising contribution regarding the simulation of jet-pipe servo-valves using ANSYS software, which could be used to accurately predict the dynamic behavior of servo-valves. Such studies, which are based on the analysis of the valve structure and physical phenomena that occur during valve operation, and testing of static and dynamic characteristics of the valve, lead to improved performance of the control system.
The article is written very systematically, with a mathematically intensive approach due to many geometric relationships and trigonometric conversions that are aimed at obtaining a detailed description of the active flow surface, the amount of flow and pressure in individual stages and the force created by pressure changes in the receiving holes that causes displacement of the valve spool.
Even though the paper may have some practical values, it needs to be supplemented by answers to some questions to qualify for publication. The main issues are as follows:
It is not entirely clear whether the identical servo valve (for which the experimental results are given) is used in the simulation and numerical analyses? Nowhere are the specifications and the type of valve that was analyzed. Only references 12 and 20 are mentioned, which are from different authors. Why was one type of valve considered in the simulation and the comparison was made with another type of valve?
How can the realized simulation model of the valve be used to create an overall dynamic model of the control system? How could a control-oriented model be obtained from the overall analysis of the servo valve, which would be suitable for controller synthesis?
How the simulation results are affected by a change in the process parameters (e.g., a change in oil viscosity, figure 8). Is this covered by the model?
The paper contains some spelling errors:
Page 1, line 28: FIG.1 --> Figure 1, line 44: [5] --> [6]
Page 2, line 49: Samashekhar [6-7] --> Samashekhar and Hiremath [6,7]
Page 4, line 141: FIG.4 --> Figure 4
Page 6, line 173: ?— The oil supply pressure --> oil density
Page 7, line 202: According to reference 18 and 19 --> [18,19]
Page 8, line 232: Formula (25): close the bracket at the end
Page 9, line 251: shown in Figure 7(a), line 254: shown in Figure 7(b) --> in the article, Figures 7(a) and 7(b) are missing !!
Page 10, line 273: 0.5mpa --> 0.5 MPa
Figure 14: Theorotical --> Theoretical
Page 13, line 343: the the pressure --> the pressure
Page 13, line 358: formulae 16, 18 and 35 --> there is no formula 35
Page ¸15, line 387: pressure is calculated using formula 18 --> using formula (17), line 388: formula 18, line 392: formula 34 --> formula 33
Page 17, line 473: PHAM --> Pham, TRAN --> Tran
